# Molnupiravir as an Early Treatment for COVID-19: A Real Life Study

**DOI:** 10.3390/pathogens11101121

**Published:** 2022-09-29

**Authors:** Michela Pontolillo, Claudio Ucciferri, Paola Borrelli, Marta Di Nicola, Jacopo Vecchiet, Katia Falasca

**Affiliations:** 1Clinic of Infectious Diseases, Department of Medicine and Science of Aging, University “G. d’Annunzio” Chieti, 66100 Pescara, Italy; 2Laboratory of Biostatistics, Department of Medical, Oral and Biotechnological Sciences, University “G. d’Annunzio” Chieti, 66100 Pescara, Italy

**Keywords:** molnupiravir, early therapy, less hospitalizations, outpatient

## Abstract

Objectives: Below we report our experience in the use of molnupiravir, the first antiviral drug against SARS-CoV-2 available to us, in the treatment of patients with COVID-19. Materials and Methods: We enrolled patients diagnosed with COVID-19 and comorbidities who were candidates for antiviral drug therapy. All patients received molnupiravir (800 mg twice daily). Blood chemistry checks were carried out at T0 and after 7/10 days after starting therapy (T1). Results: There were enrolled within the cohort 100 patients. There was 100.0% compliance with the antiviral treatment. No patient required hospitalization due to worsening of respiratory function or the appearance of serious side effects. The median downtime of viral load was ten days (IQR 8.0–13.0), regardless of the type of vaccination received. The patients who had a shorter distance from vaccination more frequently presented vomiting/diarrhea. During baseline and T1 we found significant differences in the median serum concentrations of the main parameters, in particular of platelets, RDW CV, neutrophils and lymphocytes, the eGFR, liver enzymes, as well as of the main inflammatory markers, CRP and Ferritin. Conclusion: Participants treated with molnupiravir, albeit in risk categories, demonstrated early clinical improvement, no need for hospitalization, and a low rate of adverse events.

## 1. Introduction

The new coronavirus disease 2019 (COVID-19) appeared in late December 2019 in China and quickly extended to many countries around the world. Since the appearance of COVID-19, numerous therapeutic strategies have been developed, up to the discovery of molecules with antiviral activity [1,2]. Recent oral antiviral therapies are recommended in mild/moderate cases in the presence of risk factors to avoid severe forms of COVID-19 or death [3].

In Italy, two oral antivirals have been authorized for the treatment of COVID-19 in adults who do not require supplementary oxygen therapy and who have a high risk of developing a severe form of COVID-19: PF-07321332/ritonavir (Paxlovid, Pfizer) and molnupiravir (Lagevrio, MSD). 

Paxlovid is a drug formed by the combination of nirmatrelvir (PF-07321332), a second generation protease inhibitor, and ritonavir, a drug enhancer; while Lagevrio is a RNA-dependent RNA polymerase inhibitor (RdRp), an enzyme essential for the replication of SARS-CoV-2 and that plays a pivotal role in the pathophysiology of COVID-19 [4]. Molnupiravir is an oral ribonucleoside analogue with wide spectrum antiviral activity. It is an isopropyl ester prodrug of ′Β-D-N4-hydroxycytidine (known as EIDD-1931 or NHC) and targets RdRp [5]. It stops SARS-CoV-2 replication in cell lines, infected animal models, and culture media containing airway epithelial cells and has been promoted as a treatment for COVID-19 [6,7].

While vaccination remains the best weapon to prevent COVID-19, it brings with it a few disadvantages, such as the need for booster doses, the possibility of severe allergic reactions, and unknown long-term side effects. Molnupiravir, on the other hand, presents itself as a treatment capable of allowing a reduction in symptoms and hospitalization rates if administered very early. Furthermore, the oral agent is well accepted by patients for its ease of use. One of the advantages of this target drug is that RNA-dependent polymerase has no equivalent in humans and hence there is a low risk of mutations in the human genome [7,8].

In our experience, there is an important increase in the values of the Horowitz index (P/F) with the use of molnupiravir, which is an indirect index of improvement in respiratory performance. The study also demonstrates a clinical improvement after drug treatment, which also reduces the need for hospitalization [9].

Below, we report our experience in the use of molnupiravir, the first antiviral drug against SARS-CoV-2 available to us, in the treatment of patients with COVID-19. The aim of the study is to verify the efficacy and tolerability of molnupiravir therapy in patients with early diagnosis of COVID-19 in real life, evaluating the risk of hospitalization and death after administration of the antiviral drug. The secondary endpoint is the biological response to therapy and the time to negativity of SARS-CoV-2 infection.

## 2. Materials and Methods

### 2.1. Study Design and Population

This retrospective, single-center cohort study included patients with COVID-19 admitted to the Covid Outpatient Clinic in Infectious Diseases Clinic of the University Hospital of Chieti in Abruzzo, Central Italy, from 8 January to 24 April 2022. The study was conducted in compliance with the principles embodied in the Declaration of Helsinki, and all participants provided written informed consent.

All patients were diagnosed with COVID-19 according to World Health Organization (WHO) guidelines: the presence of clinical symptoms of COVID-19 and confirmation of SARS-CoV-2 infection through a positive result on RT-PCR tests of nasopharyngeal and oropharyngeal swab specimens.

Family doctors identified patients eligible for treatment and reported them to the infectious disease specialist via a proposal form. The latter lists the data on the period of onset of symptoms, the day of positive swab, and the presence of certain risk factors.

Subsequently, having ascertained the veracity of the indications and after making telephone contact with the patients, the latter were invited to go to the Infectious Diseases Outpatient Clinic for preliminary investigations.

The inclusion criteria were patients

over the age of 18;with mild to moderate disease;with recent onset of COVID-19 (within 5 days);having comorbidities (one or more): obesity (BMI ≥ 30); chronic renal failure; uncompensated diabetes mellitus; primary or acquired immunodeficiency; oncological/hematological pathology in the active phase; cardiovascular disease; chronic pulmonary disease; chronic liver disease.

Exclusion criteria:Patients with respiratory failure needing hospitalization and oxygen therapy;Patients with incomplete information or patients transferred to other hospitals.

### 2.2. Data Collection

Epidemiological, demographic, clinical, laboratory, treatment and outcome data were extracted from medical records using a standardized data collection form. All data were checked by three physicians (J.V., C.U. and M.P), and a third investigator (K.F) adjusted any differences in interpretation between the two primary investigators.

Clinical and laboratory data were evaluated in two stages:Time 0 (T0): before the start of molnupiravir therapy;Time 1 (T1): from 7 to 10 days after the treatment.

The selected data were coded to create an anonymous dataset that included quantitative and qualitative variables. The clarifications were discussed with the coders of the epidemiology department and with the clinical teams. The analysis was conducted after the results were available for all patients.

All patients received molnupiravir (800 mg twice daily).

An arterial sample was taken in order to judge the need for oxygen therapy, which made the patient unenlistable. 

Biochemical parameters were measured in blood drawn from patients before starting antiviral therapy and after two weeks. Peripheral blood samples were collected into pre-evacuated and light-protected tubes, with no additive or with EDTA, to evaluate complete blood count and leukocyte formula, levels of high-sensitivity C-reactive protein (hs-CRP), Glycemia (Gly), aspartate aminotransferase (AST), alanine aminotransferase (ALT), gamma-glutamyl transferase (GGT), creatinemie (CRE), D-dimer, sodium (Na), potassium (K), alkaline phosphatase, lactic dehydrogenase (LDH), creatine phosphokinase (CPK), and ferritin.

In addition, other factors were evaluated, including age of the patient; date of positivization in the antigen or molecular test for SARS-CoV-2; date of onset of symptoms; distance between the date of the last vaccination and SARS-CoV-2 infection; distance between positivization and negativization; type of vaccine.

### 2.3. Statistical Analysis

Descriptive analysis was carried out using median and interquartile range (IQR) for the quantitative variables and using absolute and percentage values for the qualitative variables. The Shapiro-Wilk test indicated that quantitative data were not normally distributed. Univariate comparisons were investigated using the non-parametric Wilcoxon rank-sum test for continuous data. The non-parametric Wilcoxon matched-pairs signed-rank test was used to compare the differences between the biochemical parameters measured at the two times considered (T0 and T1). Statistical significance was taken at the <0.05 level. All analyses were performed using Stata software v17.1 (StataCorp, College Station, TX, USA).

## 3. Results

There cohort contained 100 patients. Eligible participants had mild or moderate, laboratory-confirmed COVID-19, with onset of COVID-19 signs and/or symptoms up to (and including) 5 days before treatment. The mean age was 69 ± 10.5 years (range 58.5–79), and 57% of patients were male. 

As regards natural pre-infection vaccination coverage, 85 patients of the analyzed sample were vaccinated, 68 of whom underwent a full course of vaccination, while 17 had only one or two doses. There were 15 unvaccinated patients.

In addition, 66 patients underwent vaccination with m-RNA vaccines, 1 patient underwent viral vector vaccine without a booster and 17 patients underwent heterologous vaccination. 

The most common risk factors were having cardiovascular disease (62%), obesity (36%), chronic pulmonary disease (35%), primary or secondary immunodeficiency (27%), decompensated diabetes mellitus (19%) and chronic liver disease (5%) (Table 1).

At the presentation in the outpatient clinic, the predominant symptom referred to us by the patient was cough, which involved 66 subjects in our sample; furthermore, other symptoms were frequently reported: 58 presented with fever, 43 sore throat, 56 asthenia, 43 myalgia, 23 headache, 9 dyspnea, 4 ageusia, 3 anosmia, 2 gastrointestinal symptoms and 1 patient had tachypnea (Table 2).

The median ratio of arterial oxygen partial pressure to fractional inspired oxygen (PaO_2_/FiO_2_) at admission was 392.0 (IQR 348.5–483.0) and post antiviral drug was 410.0 (IQR 376.0–445.5); this difference was found to be statistically significant (*p* < 0.0001).

There was 100.0% compliance with the antiviral treatment. No patient required hospitalization due to worsening of respiratory function or the appearance of serious side effects. Hospitalization was defined as 24 hours or longer of acute care in a hospital or similar acute care facility, including emergency rooms or facilities created to address hospitalization needs during the COVID-19 pandemic. The median downtime of viral load was ten days (IQR 8.0–13.0), regardless of the type of vaccination received.

Furthermore, we noted that there were few differences between the initial clinical presentation and the distance between the date of the last vaccine administration and the date of infection. These were found only for asthenia and gastrointestinal symptoms.

In fact, patients who had a shorter distance from vaccination more frequently presented vomiting/diarrhea (median 20 days IQR 0.0–40.0 vs. 75 days IQR 50.0–120.0, *p* = 0.037). Patients who fell ill further after vaccination had less fatigue (median 90 days IQR 60.0–120.0 vs. 60 days IQR 30.0–120.0, *p* = 0.034).

No differences were found between the type of vaccination (heterologous or not) and the development of more symptoms, nor between the latter and the number of vaccine doses received.

Table 3 shows the biochemical parameters detected between T0 and T1. During baseline and T1, we found significant differences in the median serum concentrations of the main parameters. Treatment with molnupiravir, which caused rapid resolution of SARS-CoV2 disease, was associated with a significant improvement in platelet counts, white blood cell counts, particularly neutrophils and lymphocytes, and improved liver and kidney function with eGFR, as well as in the main inflammatory markers, CRP and Ferritin.

Eighteen patients developed mild side effects following antiviral treatment: 50% had diarrhea, 16.7% headache, 11.1% nausea, 11.1% dysgeusia and 11.1% erythema of the face and upper limbs.

None of these events required medical attention or hospital admission. Patients managed to complete the entire treatment, and the adverse effects disappeared.

## 4. Discussion

Molnupiravir is a broad spectrum, directly acting oral antiviral agent that acts on the RdRp enzyme. It rivals uridine and cytidine triphosphate substrates and leads to the integration of A and G, leading to mutagenesis. Molnupiravir allows cessation of viral replication through its 2-step mutagenesis model and “error catastrophe” mechanisms [7,8].

In our cohort, the use of molnupiravir was shown to be efficacious and well tolerated. All patients completed the treatment, no patients were hospitalized, and the illness was resolved in all patients.

Our data, coming from post-therapy check-ups, allow us to attribute to molnupiravir the remission of symptoms within a week and the negativization of viral load in ten days. No patient, recalled for clinical control, showed viral rebound after the use of molnupiravir, where rebound outcomes mean COVID-19 re-infections, COVID-19–related symptoms, and hospitalizations after clinical recovery.

As seen in the literature data, as in the study by Yoseph Caraco [9], a possible clinical benefit was observed in patients who received treatment with molnupiravir at the beginning of the course of COVID-19, particularly in subjects who had manifested symptoms up to 5 days prior to treatment and who had risk factors for severe COVID-19 disease.

Oral molnupiravir in our real-life study was shown to be effective and safe in the high-risk COVID-19 population. Indeed, we observed minimal and easily manageable side effects. 

In fact, even according to several studies in the literature, treatment with all doses of molnupiravir had a profile of side effects and adverse events that was not clinically limiting during the 5 days of treatment and follow-up periods, with adverse event rates comparable to the placebo [10,11,12].

Furthermore, an important finding that emerged from our analysis is the increase in the Horowitz index (P/F) values, an indirect index of improvement in respiratory performance. The P/F index is a ratio used to evaluate lung function in patients and therefore to assess the degree of respiratory failure. It is defined as the ratio between the partial pressure of oxygen in arterial blood (PaO_2_), in millimeters of mercury, and the fraction of oxygen in the inspired air (FiO_2_). In healthy lungs, it depends on age, and it is usually between 350 and 450. What emerges from our study is an increase in this value. For the same FiO_2_ (fraction of oxygen present in the air), it results in an increase in pO_2_ dissolved in the blood and available for alveolar exchanges. Obviously, this improvement in lung function translates into a reduction in the need for oxygen therapy and also in the hospitalization rates of the most fragile COVID-19 patients.

Another result worthy of attention in our study is the increase in platelet counts, in the face of a previous thrombocytopenia in the course of SARS-CoV-2 infection. As is now known, during COVID-19 there is very often a platelet value of less than 150,000 per cubic millimeter. Several mechanisms of COVID-19–associated thrombocytopenia have been conjectured. This could be a phenomenon related to peripheral consumption, in particular endothelial loss and the formation of platelet accumulation in the lung, but bone marrow suppression and immune clearance are also possible factors [13,14]. Tachil suggests that platelets are consumed to form lung thrombi, with a possible anti-infective effect, to prevent viremic spread through the bloodstream [15].

Further information, acquired from our cohort, concerns the return to normal of the laboratory blood count RDW-CV at time T1. The red blood cell distribution width coefficient of variation (RDW-CV) is a quantitative estimate of the heterogeneity of red blood cell volume (RBC), commonly known as anisocytosis. There may be a possible association between elevated RDW and inflammation [16,17,18,19]. Inflammatory responses negatively affect red blood cell production [17,20]. Many of the pro-inflammatory cytokines upregulated in COVID-19, such as tumor necrosis factor-α and interleukin-1, can cause lowering of erythropoietin production [21]. In addition, SARS-CoV-2 infection can produce both direct injury to peripheral circulating red blood cells or erythroblasts in the bone marrow and indirect damage to red blood cells due to hemolytic anemia or intravascular coagulopathy and metabolic disorders of the iron [22]. Overall, the prevailing cause of the increase in RDW in COVID-19 is the increase in the number of older red blood cells in the circulation due to deferred clearance [18,19,23,24]. This is because older red blood cells have a reduced volume, resulting in reduced MCV. A rapid decrease in this parameter after molnupiravir therapy correlates with a clinical improvement of the patient.

Lymphocytopenia is frequently found in the early stages of SARS-CoV-2 disease. Indeed, SARS-CoV-2 can directly interfere by binding to the ACE2 receptor on the surface [25,26,27,28]. This quickly leads to lysis. Furthermore, apoptosis is also caused by the presence of a cytokine storm, characterized by markedly increased levels of interleukins (IL-6, IL-2, IL-7, granulocyte colony stimulating factor, interferon-γ-inducible protein 10, MCP-1, MIP1-alpha) and tumor necrosis factor (TNF)-alpha [28,29].

Massive cytokine activation may also be associated with the atrophy of lymphoid organs, such as the spleen, and further worsening of lymphocyte turnover [30,31]. Therefore, the percentage of lymphocytes has been suggested as a predictive biomarker for severity or recovery [32,33]. According to Han, Huang, Jiang et al., 10–12 days after the onset of symptoms, patients with a lymphocyte percentage greater than 20% can be classified as mild to moderate and have been shown to heal better and faster. Patients with less than 20% lymphocytes are classified as severe. In the period 17 to 19 days after the onset of symptoms, patients with more than 20% lymphocytes are accepted as recovering, and patients with less than 5% lymphocytes are seriously ill, with a high mortality rate [30].

In our study, we could see how the lymphocytes increased within days, thanks to early treatment with molnupiravir, indicating that the patients were in remission for symptoms and inflammation, with clinical improvement. 

Other information acquired in the context of COVID-19 is the neutropenia that characterizes the viremic or initial phase of the infection, followed by a subsequent neutrophilia in the second phase. In Huang’s study, 44 of 51 patients (86.2%) had low neutrophil counts [34,35]. As the disease progresses, it is more common to see a significant decrease in leukocyte and lymphocyte counts, accompanied by an increase in neutrophils above the normal range [30]. In our study, the initial decrease in neutrophil counts was evident, due to the intense viral replication by SARS-CoV-2. Subsequently, the number of neutrophils increases, not due to an exacerbated immune response on the part of the host, but to a return to normal in the value of neutrophils, attributed to early therapy with molnupiravir. This treatment, in fact, gave considerable support to the host’s immune response, helping it to counteract intense viral replication and thus avoiding excessive neutrophilic activation.

In the course of COVID-19, eosinophilopenia is also common. These are cells with antiviral effects [36,37,38,39]. Eosinophils contain and produce particles with antiviral activity and participate in adaptive immunity, presenting as antigen-presenting cells, as demonstrated against some respiratory viruses in vitro and in vivo, including respiratory syncytial virus and influenza [40,41].

Decreased blood eosinophil counts are positively correlated with lymphocyte counts. It has been suggested that eosinophil counts below normal levels could be a viable biomarker for diagnosing COVID-19 [42,43]. The eosinophilopenia may be the result of immune exhaustion, may represent the high rate of migration of eosinophils from peripheral blood to the infected organ or may be due to continuous and potent type 1 responses against the type 2 responses, including IL-5, which can promote life span and activation of eosinophils. Furthermore, inhibition of eosinophil output from the receptor/adhesion factors and/or induced direct apoptosis of eosinophils can be counted within these mechanisms [44,45,46].

The results of our study suggest that during COVID-19 infection the number of basophils increases, returning to a normal level. Rodriguez et al. demonstrated that basophils are depleted during acute and severe COVID-19, thus suggesting that the degree of basophil depletion may affect the efficacy of IgG responses to SARS-CoV-2 [47,48]. Li et al. found that in the early stages of the disease, basophil counts were lower in patients with COVID-19 than in controls [49,50,51]. It is known that both basophils and eosinophils can produce IL-4, which is an important cytokine for stimulating the proliferation of activated B and T lymphocytes [52,53]. Therefore, the decrease in basophil and eosinophil counts in COVID-19 patients may further explain the decrease in lymphocyte counts. 

Another typical alteration of COVID-19 is the rise in serum creatinine due to renal involvement by the virus itself.

Although COVID-19 primarily affects the lungs, it can also impact the kidneys. Acute kidney injury (AKI) was a relevant result in COVID-19 patients. There were also notable alterations in laboratory tests that indicated kidney damage, such as increased serum creatinine and urea nitrogen (BUN), proteinuria, and hematuria [54,55,56]. 

SARS-CoV-2 is known to use the angiotensin-II converting enzyme (ACEII) receptor to enter various cell lines, mainly alveolar epithelial cells but also kidneys cells [57,58]. Then, after lung infection, the virus would enter the bloodstream and accumulate in the kidneys, causing damage to renal tubular epithelial cells [57,58].

Several studies have described renal involvement mainly due to AKI, which can affect up to 70% of COVID-19 patients; the kidney is the second most affected organ, behind the lung and followed by the heart and liver [59,60,61]. 

In the literature, there have been few studies carried out on patients with renal insufficiency. On the other hand, renal clearance is not a conspicuous route of elimination for N-hydroxycytidine (NHC, active form of molnupiravir). Therefore, no dosage adjustment is required in patients with any degree of renal insufficiency. Our data demonstrated that molnupiravir can be safely administered to the patient up to an eGFR below 30. Furthermore, recovery of renal function with significant improvement in eGFR occurred after drug discontinuation.

Furthermore, renal impairment in COVID-19 also has a cytokine-induced inflammatory component that can lead to electrolyte disturbance. IL-6 can induce non-osmotic release of vasopressin and secondary hyponatremia. Another cause that can lead to increased ADH secretion among SARS-CoV-2 infected patients is a volume depletion due to digestive loss (diarrhea or vomiting). The literature reports that approximately 60% of patients with COVID-19 and watery diarrhea have moderate hyponatremia [58]. The latter would therefore represent the expression of viral replication in the intestine. In our study, there is a statistically significant difference in sodium levels, testifying to how molnupiravir, by blocking viral replication early, can affect the maintenance of electrolyte balance.

In addition to the renal point of view, the study drug proved to be safe and non-toxic in the liver. In fact, in our patient population, we could see a significant decrease in transaminases to normal after therapy, despite an initial increase caused by SARS-CoV-2 infection. It is therefore possible to understand a role of molnupiravir in the overall improvement of the patient’s health status, including in terms of liver function. Likewise, Singh et al. showed that hepatic elimination should not be a major route of elimination of NHC, and therefore hepatic insufficiency is unlikely to affect NHC exposure [59].

COVID-19 is associated with viral myositis attributable to direct invasion of myocytes or induction of autoimmune forms. COVID-19-induced myositis can range from simple myalgia to rhabdomyolysis, passing through the typical dermatomyositis. This results in an increase in enzyme markers, such as creatine kinase (CK). Virus-mediated muscle inflammation is attributed to direct ACE2 (angiotensin converting enzyme)-mediated entry and affliction of muscle fibers, leading to innate and adaptive immune activation [60,61]. In our study, there is a statistically significant difference between the two phases, with a reduction in CPK after antiviral treatment, which underlines the reduction of inflammation and viral replication, with consequent reduction of symptoms.

C-reactive protein (CRP), an acute-phase protein first described by Tillet and Francis, is synthesized by the liver in response to interleukin-6 (IL-6) and is a widely available biomarker of inflammation [62,63]. Elevated CRP concentrations are associated with cardiovascular disease and acute kidney injury (AKI) in surgical patients, with inflammatory rheumatic diseases such as rheumatoid arthritis and gout, and with incident venous thromboembolism (VTE) [64,65,66].

In our study there is a statistically significant difference between the two phases, with a reduction in CRP after antiviral treatment. In addition, in Johnson’s study of the safe population, participants who received molnupiravir had earlier and larger reductions in mean change from baseline in CRP values at all post-baseline visits than those who received a placebo [67]. 

Ferritin is a protein that stores iron; its serum level reflects the normal level of iron and helps in the diagnosis of iron deficiency anemia. The circulating ferritin level increases during viral infections and may be a marker of viral replication [68]. 

Due to its antiviral effects, in our experience, molnupiravir may have secondary anti-inflammatory effects. In fact, there was a reduction in ferritin values in the patients under study.

## 5. Conclusions 

Over the course of the pandemic, medical facilities have had to pause or limit routine, nonessential medical services and reallocate health care resources, including personal protective equipment, ventilators, and medications, to sustain the capacity to manage both patients with COVID-19 and those with other conditions requiring hospital care. Given that health care systems continue to be overburdened, decreasing the need for invasive mechanical ventilation, the need for acute care visits, and the time to hospital discharge are particularly important to preserve limited health care resources. Participants treated with molnupiravir, albeit in risk categories, demonstrated early clinical improvement, no need for hospitalization, and a low rate of adverse events. This could provide a possible opportunity for more efficient use of hospital beds.

## Figures and Tables

**Table 1 pathogens-11-01121-t001:** Characteristic demography of the study population.

100 Patients	
Age	69±10.5 years (range 58.5-79)
Gender	57% male; 43% female
Vaccination	85 Vaccinated Patients: 68 With A Full Course Of Vaccination 17 With One Or Two Doses.
	15 Unvaccinated
**Comorbidities**	
Cardiovascular Disease	62%
Obesity	36%
Chronic Pulmonary Disease	35%
Primary Or Secondary Immunodeficiency	27%
Decompensated Diabetes Mellitus	19%
Chronic Liver Disease	5%

**Table 2 pathogens-11-01121-t002:** Predominant symptoms at the presentation in the outpatient clinic.

Predominant Symptom	N
Cough	66
Fever	58
Sore Throat	43
Asthenia	56
Myalgia	43
Headache	23
Dyspnea	9
Ageusia	4
Anosmia	3
Gastrointestinal Symptoms	2
Tachypnea	1

**Table 3 pathogens-11-01121-t003:** Biochemical parameters between T0 and T1.

	T0	T1	*p*-Value *
GB (10^6^/mmc)	4.6 (4.1–5.0)	4.5 (4.1–4.9)	0.130
HB (g/dL)	13.6 (12.1–14.6)	13.3 (12.3–14.4)	0.059
HCT (%)	41.7 (37.5–44.5)	41.1 (37.6–43.6)	0.054
MCV (fL)	90.2 (87.1–93.3)	89.9 (86.7–92.6)	0.27
MCH (pg)	29.7 (27.9–30.6)	30.3 (28.2–30.8)	0.766
MCHC (g/dL)	32.6 (31.9–33.3)	32.8 (32.0–33.5)	0.633
PLT (10^3^/mmc)	196.0 (157.0–234.0)	238.0 (200.0–290.0)	<0.0001
RDWCV (%)	13.5 (12.8–15.0)	13.3 (12.7–14.6)	<0.0001
MPV (fL)	10.6 (10.2–11.3)	10.7 (10.1–11.3)	0.939
WBC (10^3^/µL)	6.0 (4.7–7.5)	7.6 (5.7–9.5)	<0.0001
NEUTROPHILS (10^3^/µL)	3.2 (2.4–4.9)	4.7 (3.4–5.9)	<0.0001
LYMPHOCYTES (10^3^/µL)	1.5 (1.0–2.1)	1.8 (1.4–2.4)	<0.0001
MONOCYTES (10^3^/µL)	0.6 (0.5–0.8)	0.6 (0.5–0.8)	0.368
EOSINOPHILS (10^3^/µL)	0.1 (0.0–0.1)	0.1 (0.1–0.2)	<0.0001
BASOPHILS (10^3^/µL)	0.0 (0.0–0.0)	0.0 (0.0–0.1)	<0.0001
D-DIMER (mg/L)	0.5 (0.3–1.0)	0.5 (0.3–1.0)	0.073
GLYCEMIA (mg/dL)	103.0 (93.0–126.0)	97.0 (88.0–120.0)	0.104
GLOMERULAR FILTRATE (ml/min)	77.0 (62.8–88.2)	81.0 (65.3–92.4)	<0.0001
Na (mmol/L)	140.0 (138.0–141.0)	140.5 (140.0–141.0)	<0.0001
K (mmol/L)	4.2 (3.9–4.4)	4.2 (4.0–4.4)	0.057
AST(U/L)	22.0 (18.0–28.0)	19.0 (16.0–23.0)	<0.0001
ALT (U/L)	21.0 (15.0–32.0)	18.0 (13.0–26.0)	<0.0001
GGT (U/L)	25.0 (18.5–37.5)	23.5 (17.0–35.0)	0.140
ALKALINE PHOSPHATASE (U/L)	63.5 (51.0–79.5)	62.0 (50.0–79.0)	0.150
LDH (U/L)	175.0 (152.0–200.0)	172.0 (150.0–193.0)	0.596
CPK (mg/mL)	66.5 (42.0–110.0)	58.5 (42.0–96.5)	0.010
CRP (mg/L)	14.2 (7.2–26.1)	3.0 (1.7–7.2)	<0.0001
FERRITIN	132.6 (71.7–254.4)	102.2 (50.9–192.4)	<0.0001

Data are expressed as median and interquartile range (IQR); * Wilcoxon matched-pairs signed-rank test.

## Data Availability

The data presented in this study are available on request to the corresponding author, K.F.

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
