# Peer review of "Molnupiravir as an Early Treatment for COVID-19: A Real Life Study"

_pathogens, 2022, doi:10.3390/pathogens11101121_

Round 1

Reviewer 1 Report

The research objective is very noble and will be helpful for a large group of readers. However, to make it more understandable for the readers it's necessary to define the importance of the drug. The conclusions made by the authors must be supported by sufficient data and discussed adequately.

1. The authors should rewrite the introduction section and be describing the importance of this research adequately and how the particular biochemical test is the best way to show the effect of the drug.

2. The results must be convincing and statistically significant. Show all the supporting data and refer to it while describing it in the result section.

3. The discussion section needs to be rewritten. All the findings must be supported with references. 

4. Need to improve the English of the manuscript.

Author Response

The research objective is very noble and will be helpful for a large group of readers. However, to make it more understandable for the readers it's necessary to define the importance of the drug. The conclusions made by the authors must be supported by sufficient data and discussed adequately.

  1. The authors should rewrite the introduction section and be describing the importance of this research adequately and how the particular biochemical test is the best way to show the effect of the drug.

We have rewritten and expanded the introduction

  1. The results must be convincing and statistically significant. Show all the supporting data and refer to it while describing it in the result section.

We have better explained the significant data in the results

  1. The discussion section needs to be rewritten. All the findings must be supported with references. 

We have rewritten

  1. Need to improve the English of the manuscript.

We have improved the English of the manuscript

Reviewer 2 Report

With this study, authors share their experience for use of monlupiravir, the RNA-dependent RNA polymerase inhibitor which is a key enzyme for the replication of SARS-CoV-2. The aim of the study is to verify the efficacy and tolerability of monlupiravir therapy in patients with early diagnosis of COVID-19 in real-life, evaluating the risk of hospitalization and death after administration of the antiviral drug. 

The observation importantly indicate the increase in the Horowitz index (P/F) values with the use of monlupiravir, which is an indirect index of improvement in respiratory performance. The study also demonstrates clinical improvement after the drug treatment which also waives the need of hospitalization. 

Authors are advised to comment on any viral rebound after the monlupiravir use.

Author Response

  1. Referee

With this study, authors share their experience for use of monlupiravir, the RNA-dependent RNA polymerase inhibitor which is a key enzyme for the replication of SARS-CoV-2. The aim of the study is to verify the efficacy and tolerability of monlupiravir therapy in patients with early diagnosis of COVID-19 in real-life, evaluating the risk of hospitalization and death after administration of the antiviral drug. 

The observation importantly indicate the increase in the Horowitz index (P/F) values with the use of monlupiravir, which is an indirect index of improvement in respiratory performance. The study also demonstrates clinical improvement after the drug treatment which also waives the need of hospitalization. 

Authors are advised to comment on any viral rebound after the monlupiravir use.

We did not observed viral rebound after the molnupiravir use

Reviewer 3 Report

In this study, the authors reported their experience in the use of molnupiravir in the treatment of patients with COVID-19. They concluded that participants treated with molnupiravir, albeit in risk categories, demonstrated early clinical improvement, no need for hospitalization, and a low rate of adverse events.

 Several suggestions:

1.     It is better to have another table to summarize the demography of the study population, such as age, gender, comorbidity, vaccination, etc. [written in lines 120-146].

2.     Title, [Monlupiravir] needs change to [Molnupiravir].

3.     Line 41, [Both are RNA-dependent RNA polymerase inhibitors (RdRp)], actually, Paxlovid is a protease inhibitor. Please check!

4.     Line 63, [RT tests -PCR] needs change to [RT-PCR tests].

5.     When [coronavirus disease 2019 (COVID-19)] was mentioned in line 31, use COVID-19 thereafter, not [coronavirus disease 2019 (COVID-19)] in lines 37-38, line 312, etc. Also in SARS-CoV-2, full-name was written at its first appearance, then use SARS-CoV-2 thereafter, not [Sars-Cov-2] in lines 107, 108, or line 311, etc.

6.     Line 125, [68 of whom underwent a full course of vaccination, while 15 had only one or two doses.] 68+15=83, not 85, please check.

7.     Please add a reference after the paragraph or sentence in lines 173, 194, 264.

8.     There is no [placebo] group in this study. It is better to cite a reference of patients with COVID-19 but no treatment. What are their outcomes? i.e. percentages of severe symptoms, hospitalization, etc. for a comparison.

Author Response

3.Referee

In this study, the authors reported their experience in the use of molnupiravir in the treatment of patients with COVID-19. They concluded that participants treated with molnupiravir, albeit in risk categories, demonstrated early clinical improvement, no need for hospitalization, and a low rate of adverse events.

 Several suggestions:

  1. It is better to have another table to summarize the demography of the study population, such as age, gender, comorbidity, vaccination, etc. [written in lines 120-146].

We added tables 1 and 2

  1. Title, [Monlupiravir] needs change to [Molnupiravir].

We corrected

  1. Line 41, [Both are RNA-dependent RNA polymerase inhibitors (RdRp)], actually, Paxlovid is a protease inhibitor. Please check!

We corrected

  1. Line 63, [RT tests -PCR] needs change to [RT-PCR tests].

We corrected

  1. When [coronavirus disease 2019 (COVID-19)] was mentioned in line 31, use COVID-19 thereafter, not [coronavirus disease 2019 (COVID-19)] in lines 37-38, line 312, etc. Also in SARS-CoV-2, full-name was written at its first appearance, then use SARS-CoV-2 thereafter, not [Sars-Cov-2] in lines 107, 108, or line 311, etc.

We corrected

  1. Line 125, [68 of whom underwent a full course of vaccination, while 15 had only one or two doses.] 68+15=83, not 85, please check.

We corrected

  1. Please add a reference after the paragraph or sentence in lines 173, 194, 264.

We added

  1. There is no [placebo] group in this study. It is better to cite a reference of patients with COVID-19 but no treatment. What are their outcomes? i.e. percentages of severe symptoms, hospitalization, etc. for a comparison.

We have rewritten

Round 2

Reviewer 1 Report

The authors have improved the manuscript.

Reviewer 3 Report

The authors have addressed the issues I raised previously in this revised manuscript.